# Extracting quantitative dielectric properties from pump-probe spectroscopy

Arjun Ashoka[1], Ronnie R. Tamming[2,3,4], Aswathy V. Girija [1], Hope Bretscher [1], Sachin Dev Verma [1,7], Shang-Da Yang[5], Chih-Hsuan Lu[5], Justin M. Hodgkiss[3,4], David Ritchie [1], Chong Chen[1], Charles G. Smith[1], Christoph Schnedermann [1], Michael B. Price [3,4], Kai Chen[2,4,6] & Akshay Rao [1✉]

Optical pump-probe spectroscopy is a powerful tool for the study of non-equilibrium electronic dynamics and finds wide applications across a range of fields, from physics and chemistry to material science and biology. However, a shortcoming of conventional pump-probe spectroscopy is that photoinduced changes in transmission, reflection and scattering can simultaneously contribute to the measured differential spectra, leading to ambiguities in assigning the origin of spectral signatures and ruling out quantitative interpretation of the spectra. Ideally, these methods would measure the underlying dielectric function (or the complex refractive index) which would then directly provide quantitative information on the transient excited state dynamics free of these ambiguities. Here we present and test a model independent route to transform differential transmission or reflection spectra, measured via conventional optical pump-probe spectroscopy, to changes in the quantitative transient dielectric function. We benchmark this method against changes in the real refractive index measured using time-resolved Frequency Domain Interferometry in prototypical inorganic and organic semiconductor films. Our methodology can be applied to existing and future pump-probe data sets, allowing for an unambiguous and quantitative characterisation of the transient photoexcited spectra of materials. This in turn will accelerate the adoption of pump-probe spectroscopy as a facile and robust materials characterisation and screening tool.

[1] Cavendish Laboratory, University of Cambridge, J.J. Thomson Avenue, CB3 0HE Cambridge, UK. [2] Robinson Research Institute, Faculty of Engineering, Victoria University of Wellington, Wellington 6012, New Zealand. [3] School of Chemical and Physical Sciences, Victoria University of Wellington, Wellington 6012, New Zealand. [4] MacDiarmid Institute for Advanced Materials and Nanotechnology, Wellington 6012, New Zealand. [5] Institute of Photonics Technologies, National Tsing Hua University, Hsinchu 30013, Taiwan. [6] The Dodd-Walls Centre for Photonic and Quantum Technologies, Dunedin 9016, New Zealand. [7] Present address: Department of Chemistry, Indian Institute of Science Education and Research Bhopal, Bhopal Bypass Road, Bhopal 462066 Madhya Pradesh, India. ✉email: ar525@cam.ac.uk

Optical pump-probe spectroscopy has historically proven itself to be an excellent tool with which to study the non-equilibrium photophysics of a range of materials and is being increasingly employed for material optimisation and screening. The advantage of this technique is the broad range of timescales that it can resolve, ranging from femtoseconds to seconds. Recently, it has deepened our understanding of (sub-) picosecond exciton dynamics in organic semiconductors, hot carrier cooling, and recombination dynamics in inorganic–organic metal halide perovskites and excitonic effects in two-dimensional semiconductors[1–3]. On slower timescales, it has also provided critical insights into (micro-)second charge carrier dynamics for photocalatytic conversion reactions[4,5].

One of the most common experimental realisations of pump-probe spectroscopy is to optically photoexcite the system of interest with a pump pulse and stroboscopically read out changes in the transmission with a time-delayed optical probe pulse. Changes in the probe pulse spectrum are then typically assigned to different excitations in the material from which the photophysical identities and dynamics are inferred. This direct interpretation of pump-probe spectra stems from the fact that in the case of dilute solutions, assuming the Beer-Lambert law, the change in absorbance is usually computed from the differential transmission as,

$$\triangle \alpha = -\log_{10}\left[\frac{\triangle T}{T} + 1\right] \qquad (1)$$

where $\triangle T$ corresponds to the change in probe transmittance upon photoexcitation. In thin film materials, however, measuring isolated changes in the transmission and reflection only serve as incomplete proxies for the transient absorption, as both transmission and reflection contribute to the obtained signal[6]. This raises an important challenge as changes in the transmission/reflection and absorption are no longer exactly equivalent and a direct interpretation of the differential spectra is obscured.

For example, it has recently been argued that the dispersive transient transmission spectral features seen near the bandgap of metal halide perovskites arise not as optical signatures of underlying excited states, but instead from changes in the real part of the refractive index which contribute to changes in the reflection[6]. In fact, frequency domain interferometry (FDI) studies on $CsPbBr_3$, which exclusively extract changes arising from the real refractive index, have confirmed that changes in the refractive index can be significant[7]. On the other hand, photo-induced reflection/refractive index effects are hardly ever discussed in other important optoelectronic materials, such as organic semiconductor films. In these systems, Eq. 1 is often employed despite the fact that a transfer matrix type approach is always better suited to describing the spectra, as a thin film geometry exacerbates the incompatibility of Beer–Lambert Law with Maxwell's Equations[8].

Initial approaches to mitigate these ambiguities in studying differential spectra have considered measuring both the differential transmission and reflection, multiangle reflectance, or the optical ellipticity from which the true photoexcited transient absorption changes can be calculated[9,10]. These methods are, however, experimentally challenging on the commonly studied thin-film materials and not universally applicable, especially on solution processed thin-film semiconductors where morphological microstructures complicate the analysis[11]. Further, while the absorbance $\alpha$ is a good measure of the population and oscillator strengths of excited states in a material, it is also inversely related to the real refractive index $n$, which can vary significantly upon photoexcitation, complicating the analysis.

Here, we present and benchmark a model-independent and universally applicable route to transform broadband differential optical spectra to changes in the real and imaginary part of the dielectric function from which changes in the true underlying excitation spectrum of the material can be confidently assigned. We make use of the well-established Kramers–Kronig (KK) relations and their variational counterpart[12–14]. As shown schematically in Fig. 1, the inputs to this analysis are only the experimentally measured differential pump probe spectrum and the material's static spectra (transmission, reflection, and/or ellipsometry), making it straightforward to implement on data being taken with most current pump-probe methods, or indeed data already taken, with no changes in instrumentation required.

## Results

To quantitively characterise the optical signatures in these experiments, knowledge of the photoexcited population and oscillator strength of the probed transition is critical. These

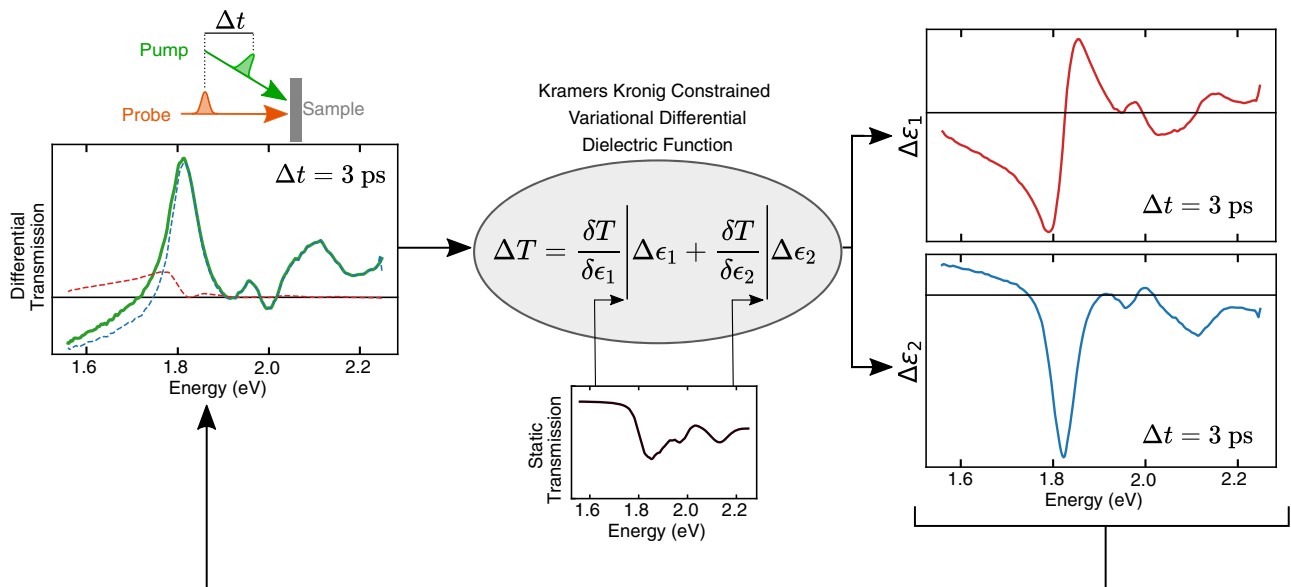

**Fig. 1 Schematic of the Kramers–Kronig based analysis technique.** Here, equipped with the differential spectra of a pump probe experiment (green) and the static spectrum (black), one can extract quantitative changes in the transient complex dielectric function (red and blue).

quantities are represented by the photoexcited joint-density-of-states and the transition dipole matrix elements, respectively. These two properties are embedded in the imaginary part of the transverse dielectric function $\widetilde{\varepsilon}$, i.e., $\varepsilon_2$, where $\widetilde{\varepsilon} = \varepsilon_1 + i\varepsilon_2$.

For a material subject to an oscillating transverse electromagnetic field, using Fermi's golden rule and the macroscopic power dissipation theorem, $\varepsilon_2$ is given as,

$$\varepsilon_2(\omega) = \frac{2\pi}{N_e m_e} \sum_{i \in GS} \frac{\omega_p^2}{\omega_i^2} \sum_T \underbrace{|p_{T_i}|^2}_{\text{Oscillator Strength}} \underbrace{\delta(E_{T_i} - \hbar\omega)}_{\text{Joint Density of States (JDOS)}}$$

(2)

where the real part of the dielectric function can be obtained by the KK transform,

$$\varepsilon_1(\omega) = \frac{1}{\pi}\wp \int_{-\infty}^{\infty} \frac{\varepsilon_2(\omega')}{\omega' - \omega} d\omega'.$$

(3)

Here, $m_e$ is the electron mass, $N_e$ is the number of electrons in molecular or crystal unit cell, $\omega_p$ is the plasma frequency, $\omega$ the photon frequency, and the factor of 2 accounts for the spin degeneracy. The first sum (over $i$) extends over the available states in the ground state (GS) manifold (or equivalently the Brillouin Zone for a crystal) and the second sum is over the available transitions (T). The matrix elements of the momentum operator between the wavefunctions in the ground and excited (or valance and conduction bands for a crystal) is given as $p_{T_i}$ with an associated transition energy $E_{T_i}$. The matrix elements of the momentum operator can be recast as the electric dipole operator that represents the oscillator strength of the transition. The Dirac delta function represents the existence of states in the ground and excited state manifold that would allow for a transition at an energy $\hbar\omega$, which is referred to as the joint density of states (JDOS). This picture is naturally modified for excitonic systems, but the overall dependence of $\varepsilon_2$ on the JDOS and the dipole matrix elements remains universal[15].

The approach we present exploits the mathematical relationship between the imaginary part of the dielectric function $\varepsilon_2(\omega)$ and any optical spectrum of interest, for example using the well-known relations between the transmission, reflection, and absorption for a thin film sample or a semiconductor wafer (see SI 2). In general, the transmission spectrum of a material is a scalar function of energy, which is parameterised by the complex dielectric function, $T(\widetilde{\varepsilon}(\omega))$. This relationship can be re-parameterised by the real and imaginary dielectric functions of energy, $\varepsilon_1(\omega)$ and $\varepsilon_2(\omega)$, which are KK constrained, i.e., $T(\varepsilon_1, \varepsilon_2)$ where $\varepsilon_1$ and $\varepsilon_2$ are related by Eq. 3.

When a material is photoexcited, its JDOS is transiently modified. This imprints itself on the imaginary part of the dielectric function $\varepsilon_2(\omega)$ as two typical features. First, a photobleaching of a ground state transition, which would manifest as a reduction in the JDOS at the original transition energy and consequently a negative $\triangle\varepsilon_2(\omega)$ (pump$_{\text{on}}$-pump$_{\text{off}}$). Second, a photo-induced absorption which would manifest as an augmented JDOS with new transitions and consequently a positive $\triangle\varepsilon_2(\omega)$. These changes in the JDOS are weighted by their electronic transition dipole matrix elements (oscillator strengths) of either the ground state (photobleaching) or the excited state transitions (photoinduced absorption). $\triangle\varepsilon_2(\omega)$ therefore contains all the quantitative information of the transient carrier dynamics and their transition probabilities. We note that this formalism does not include stimulated emission (SE) features as net transmission changes that result from radiation from the material rather than changes in optical constants do not measure the transient photoexcited JDOS as they do not appear in Eq. 2. However, with appropriate interpretation of spectral photobleaching features in $\triangle\varepsilon_2(\omega)$, based on the magnitude of the Stokes shift, one can deconvolve the SE band from the true transient JDOS (SI 1).

As the KK transform holds for any material and dielectric function, $\triangle\widetilde{\varepsilon} = \triangle\varepsilon_1 + i\triangle\varepsilon_2$ can be thought of as the difference between the excited state and ground state dielectric functions. $\triangle\widetilde{\varepsilon}$ is therefore also KK constrained, i.e., knowing only $\triangle\varepsilon_2$ across the whole frequency range allows one to determine $\triangle\varepsilon_1$ over the same range. We note that while usually the KK relations are not strictly obeyed in third-order spectroscopies, due to the strict time ordering of the pump and probe pulse, the KK relation can be applied to the probe spectrum. A well-established complication arising from the use of the KK relation on the ground state dielectric function is that experimental limitations truncate the frequency range that can be accessed which causes the use of the KK relations on the measured dielectric properties to become less exact. The effect of transitions a distance $\omega - \omega_0$ from a transition at $\omega_0$ are, however, suppressed by the term $1/(\omega - \omega_0)$ in the KK relations (Eq. 3) and hence, while non-locality is an ever-present issue with the use of the KK relations, they can still be applied over finite experimental bandwidths[16]. In typical semiconducting materials, however, $\triangle\widetilde{\varepsilon}(\omega)$ is usually an energetically local perturbation to the dielectric function, and hence most of the features, especially in the weak photoexcitation limit, are also energetically local and the KK constraint can be accurately applied over the narrower energy range where the spectral features are present.

As the static transmission spectrum, $T$, is parameterised by the real and imaginary part of the dielectric function, in the spirit of the differential dielectric function, we can take a Taylor expansion which yields normalised changes in the transmission spectra to first order as,

$$\frac{\triangle T}{T} = \frac{1}{T}\left[\left(\frac{\partial T}{\partial \varepsilon_1}\right)\triangle\varepsilon_1 + \left(\frac{\partial T}{\partial \varepsilon_2}\right)\triangle\varepsilon_2\right].$$

(4)

Here the derivatives are given by the static transmission function $T(\varepsilon_1, \varepsilon_2)$ and $\triangle\varepsilon_{1,2}$ represents the differential dielectric function discussed above (Fig. 1).

We use the functional form of $T(\varepsilon_1, \varepsilon_2)$ for a bifacial thin-film geometry using Fresnels Equations (see SI 2). We note that for more complex sample structures including interferences and Fabry–Pérot resonances, $T(\varepsilon_1, \varepsilon_2)$ can be efficiently calculated using transfer matrix modelling[17,18]. Once $T(\varepsilon_1, \varepsilon_2)$ has been parameterised, it is possible to calculate the derivatives $\left(\frac{\partial T}{\partial \varepsilon_1}\right)$ and $\left(\frac{\partial T}{\partial \varepsilon_2}\right)$ analytically as a function of $\varepsilon_1$ and $\varepsilon_2$. In order to compute these value of derivatives for a given experimental spectrum, it is first necessary to accurately fit the static transmission to a function of $\varepsilon_{1,2}$, i.e., to transform $T(\omega)$ to $T(\widetilde{\varepsilon}(\omega))$ so that the derivatives can be calculated from their analytic form. A robust technique recently developed is to employ a variational dielectric function to exactly transform the data by fitting it with a large set of analytically KK-constrained functions (such as Lorentzians) which constitute the dielectric function[14]. The procedure is to first fit the data to a set of few Lorentz oscillators that capture the bulk of the spectral weight and to then fix these oscillators and fit the residues to a KK-constrained variational dielectric function that consists of a large fixed energetic grid (typically less than half the experimental dataset size) of oscillators where only the amplitudes are varied. The selection of analytic functions that can serve as variational oscillators is discussed in[14]. Once the ground state static transmission has been described by $\varepsilon_1$ and $\varepsilon_2$ and its partial derivatives computed analytically, we can evaluate the pump-probe spectra. Here we fit the differential spectra $\frac{\triangle T}{T}$ to Eq. 4, using the previously computed derivatives in a similar manner

described above, to determine the relative spectral contributions of $\triangle \varepsilon_1$ and $\triangle \varepsilon_2$ (Fig. 1, red, blue lines respectively).

It is important to note that using such a large number of oscillators to fit the spectra and its derivatives to retrieve $\widetilde{\varepsilon}$ and $\triangle \widetilde{\varepsilon}$ disallows any physical interpretation of each of the oscillators as particular electronic excitations or species. It allows us, however, to move between measured spectra and the underlying dielectric function in a model-independent way where no assumptions about the underlying spectral carrier distributions, and their line shapes have been made[14].

**Transient transmission of a CsPbBr₃ thin film**. To illustrate this methodology, we have applied it to the excited state transient transmission spectrum of a microcrystalline inorganic metal halide perovskite CsPbBr₃ film published earlier[7]. Metal-halide perovskite thin films have recently garnered much interest as promising optoelectronic materials due to their high photoluminescence quantum efficiencies and power conversion efficiencies in light emitting diodes and photovoltaic cells[19–21]. To develop improved perovskite materials, a detailed understanding of loss channels is required. Here, pump-probe spectroscopy has been extensively used to quantify long and short time dynamics after photoexcitation. However, at short times, spectral signatures display a dispersive line shape, which has been ascribed to strong signal distortion due to the real part of refractive index.

As shown in Fig. 2a, b, the method we present is able to combine transient transmission measurements and static ellipsometry and transmission measurements reported in[22,23] to quantitively deconvolute the contributions to the differential

transmission spectra from both the real and imaginary part of the dielectric function (SI 3). We find that the contribution from the derivative line shape $\triangle \varepsilon_1$ to the overall transient transmission are smaller than expected given previous discussions in the literature on the impact of refractive index changes, which would be modulated by $\triangle \varepsilon_1$. Consequently, the transient transmission spectrum in these materials is most sensitive to $\triangle \varepsilon_2$. We can further calculate the expected changes in the surface reflection as shown in Fig. 2c and quantitively extract the transient dielectric function. We note that the spectral shape of the transient transmission follows the change in $\triangle \varepsilon_2$ closely, as expected, with small but significant deviations below the bandgap.

In order to benchmark our approach, we compare our calculated $\triangle n$ spectrum for CsPbBr₃ perovskite with experimental data previously obtained by FDI, which experimentally determined the change in the real refractive index $\triangle n$ at the same photoexcitation density. The details of the FDI experiment are discussed in the "Methods" section and in[7]. We obtain the transient complex refractive index from the transient dielectric function using the partial derivative of $\widetilde{n} = n + ik = \sqrt{\varepsilon}$,

$$\triangle \widetilde{n} = \triangle n + i\triangle k = \frac{1}{2\sqrt{\widetilde{\varepsilon}}}\triangle \widetilde{\varepsilon}, \qquad (5)$$

where $n$ is the real refractive index and $k$ is the extinction coefficient. As illustrated in Fig. 2d, we find quantitative agreement between our calculated $\triangle n(\omega)$ spectra obtained via variational KK analysis (solid lines) and the measured spectra (open circles), validating our approach and the use of the KK relations on a relatively energetically narrow spectrum. We find

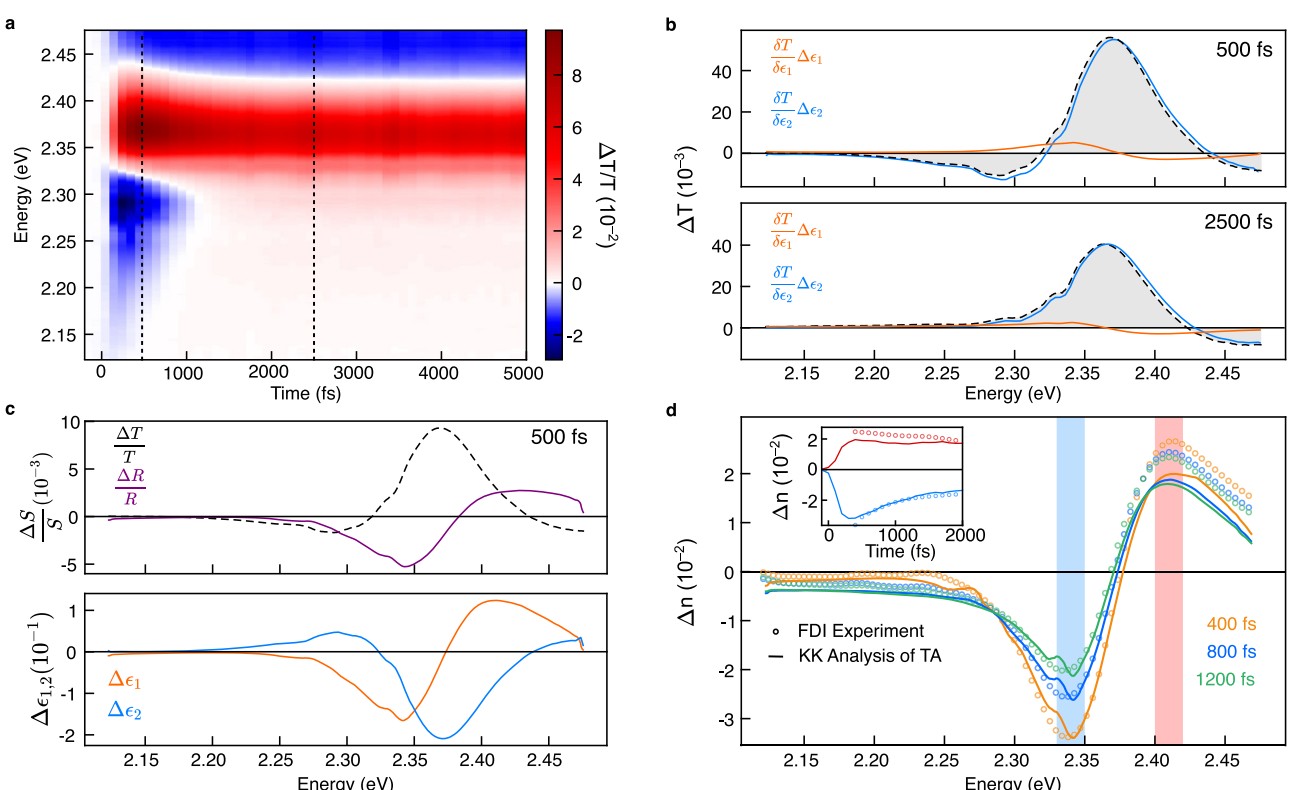

**Fig. 2 Extraction and benchmarking of the picosecond transient dielectric function of CsPbBr3. a** Broadband transient transmission spectrum of CsPbBr₃. **b** Contribution of changes in the real and imaginary part of the dielectric function to the measured transient transmission at 500 and 2500 fs, demonstrating that transient transmission is a very sensitive measure of changes in the underlying imaginary dielectric function. **c** The transient dielectric function and normalised transient spectra S(ω) computed from the measured differential transmission at 500 fs demonstrating that neither the transient reflection and transmission capture the spectral information of $\triangle \varepsilon_2$ exactly. **d** Comparison of the extracted $\triangle n(\omega)$ with that retrieved from a time-resolved frequency domain interferometry (FDI) experiment on the same sample at the same excitation density demonstrating quantitative agreement. Inset: Comparison of $\triangle n(\omega)$ kinetics over the two spectral bands indicated.

that photoinduced changes in the real refractive index, $n$, are indeed large and evolve in both spectral position and intensity over the first 500 fs. $\Delta n(\omega)$ is however not strongly imprinted on the transient transmission spectrum (SI 4). We can therefore conclude that the spectral features observed at early times near the bandgap of $CsPbBr_3$ in a transmission pump-probe experiment can confidently be assigned to changes in $\Delta\varepsilon_2$.

**Transient transmission of a pentacene thin film.** To bolster our understanding of the effect of refractive index changes and reflection contributions to the transient transmission spectra on different semiconducting systems we performed the same analysis and benchmark experiments on an evaporated film of the prototypical organic semiconductor pentacene. This system offers a challenge to this technique due to the large number of congested spectral features present in its transient spectra. Pentacene has recently gathered much attention in the field of optoelectronics as it has the ability to perform efficient singlet fission, i.e., generate two excitations of half the energy of the optical excitation with near unity efficiency, paving the way to potential high efficiency photon management in photovoltaics[24,25]. Pump-probe spectroscopy has played a key role in understanding the photophysical dynamics of this material[26].

As shown in Fig. 3a we performed broadband femtosecond pump-probe spectroscopy photoexciting the sample with a with sub 20 fs, broadband pump centred at 2.25 eV (for details see Methods). We observe the expected singlet fission dynamics over the first 50 fs, with conversion between spectral features of the singlet and triplet at 1.81 eV and below 1.70 eV respectively, in

agreement with previous reports[27]. Upon applying the KK-based analysis for this material using ellipsometry data reported in[28] and measured transmission data (SI 5), we find the contribution to the measured differential transmission from the changes in $\Delta\varepsilon_1$ are significant only on the low energy, sub-bandgap feature (Fig. 3b, orange). This analysis suggests that either the sub-bandgap triplet population and/or transition dipoles are larger than one would have otherwise estimated from the transient transmission alone. As shown in Fig. 3c we are able to quantify changes in the transient dielectric function and compute the expected transient reflection spectrum.

As illustrated in Fig. 3d, we benchmarked the variational KK analysis (solid lines) against FDI experiments (open circles) and again find excellent agreement over the whole spectral range. This demonstrates that the variational KK analysis can be applied on congested pump-probe spectra and achieves quantitative precision on the signal shape and peak to peak ratios further validating our approach and the use of the KK relations on a relatively energetically narrow spectrum.

**Transient reflection on a GaAs wafer.** Finally in order to demonstrate the efficacy of this technique on different experimental geometries and spectra, we carried out ultrafast transient reflection measurements (pump centred at 1.77 eV, 15 fs, for details see Methods) on a GaAs wafer, which cannot be measured in transmission, and performed the same KK analysis (SI 6). GaAs is a well-understood material that currently holds the efficiency record in power conversion efficiency for photovoltaic systems and sets the benchmark for high quality semiconducting

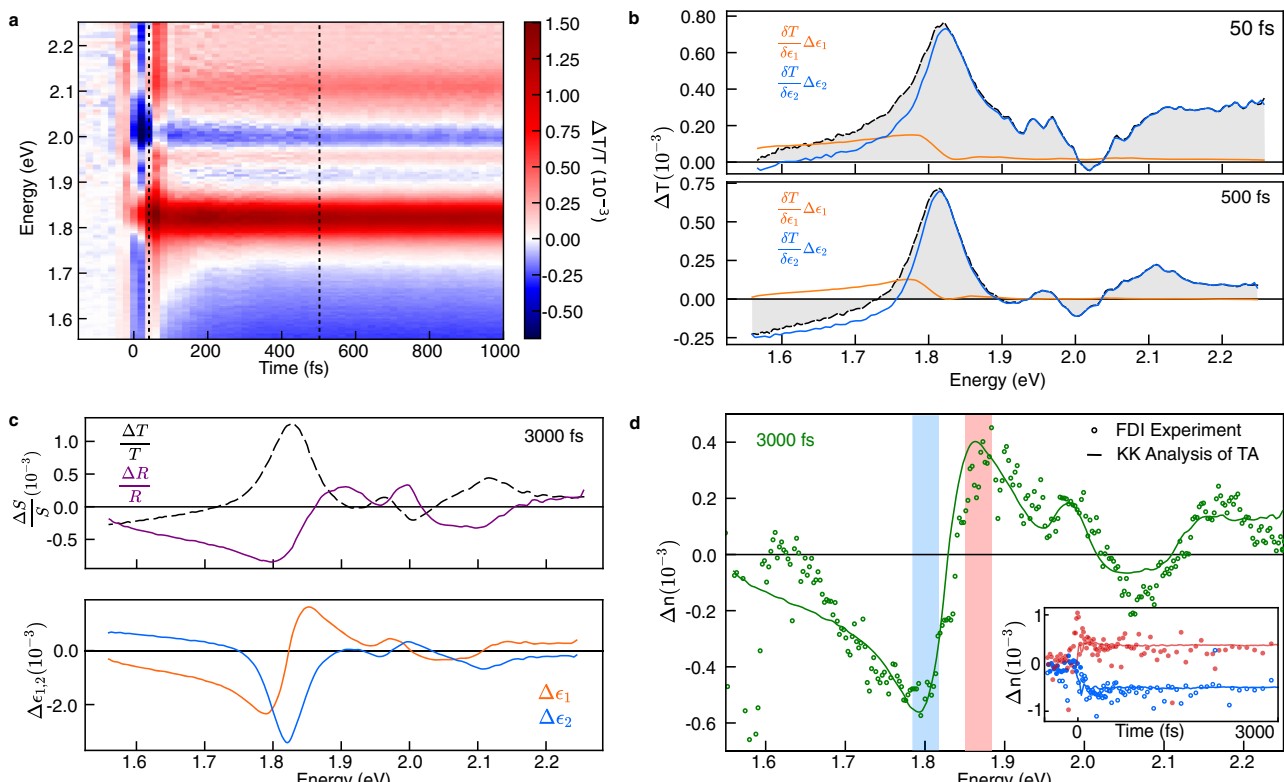

**Fig. 3 Extraction and benchmarking of the femtosecond transient dielectric function of Pentacene. a** Broadband transient transmission spectrum of pentacene. **b** Contribution of changes in the real and imaginary part of the dielectric function to the measured transient transmission at 25 and 500 fs, demonstrating that transient transmission on the low energy feature has significant contributions from the real part of the dielectric function. **c** The transient dielectric function and normalised transient spectra $S(\omega)$ computed from the measured differential transmission at 3000 fs demonstrating that neither the transient reflection and transmission capture the spectral information of $\Delta\varepsilon_2$ exactly. **d** Comparison of the extracted $\Delta n(\omega)$ with that retrieved from a time-resolved frequency domain interferometry (FDI) experiment on the same sample. Inset: Comparison of $\Delta n(\omega)$ kinetics over the two spectral bands indicated.

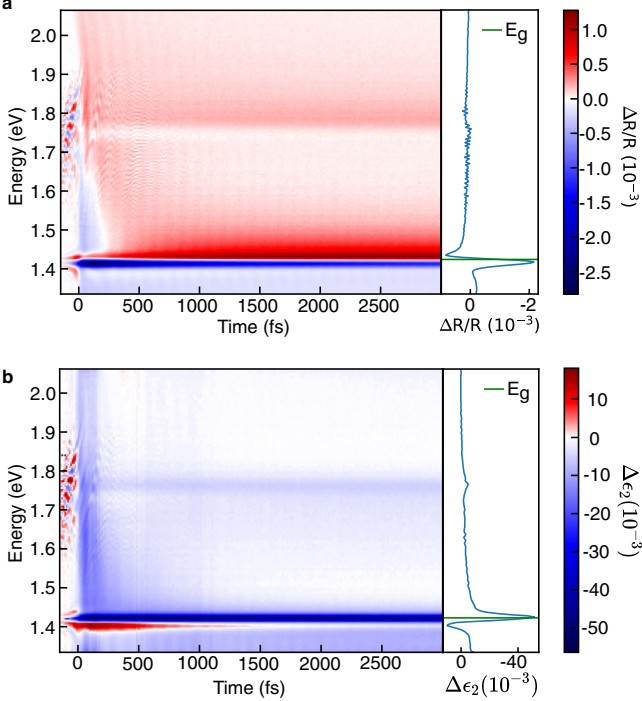

**Fig. 4 Interpretation of the transient reflection response of a GaAs wafer.** **a** Broadband transient reflection map and spectral cut at 500 fs (blue line) indicating that the transient spectral features are unconventional if interpreted as the transient absorption with the bandgap not lying on any of the spectral features. **b** Broadband $\Delta\varepsilon_2$ extracted using the KK analysis and a spectral cut at 500 fs (blue line) showing that the broadband spectra now resembles that of a typical inorganic semiconductor with the photobleach of $\Delta\varepsilon_2$ occurring at the bandgap. The green line in the side panels indicates the bandgap of GaAs.

properties[29]. Quantitative interpretation of its broadband spectra and photophysical dynamics would allow for direct comparison with other emerging thin film solar energy harvesting materials, potentially hinting at ways to improve their photovoltaic efficiency. For instance, for a given photoexcitation density, quantitative comparison of $\Delta\varepsilon_2(\omega, t)$ between a state-of-the-art inorganic-organic perovskite and a high quality GaAs wafer could offer insights into differences in non-radiative carrier decay pathways between the two systems.

An immediate impediment to such a quantitative comparison is that the transient reflection spectra of GaAs and the transient transmission spectra of thin-film semiconductors are spectrally dissimilar. As shown in Fig. 4a, the transient reflection spectra of GaAs has spectral features that are not straightforward to assign without further analysis of the spectra, unlike the transient transmission spectra of CsPbBr$_3$ (Fig. 2a). For instance, the observed photo bleach-like signal at 1.417 eV cannot be assigned to any of the spectral peaks known for GaAs, and we identify a spectral peak above the known bandgap at 1.435 eV.

Upon performing the KK analysis, however, we are able to recover changes in $\Delta\varepsilon_2$ and find a negative spectral peak at the expected bandgap value of 1.424 eV[30]. Further, we see a clear subbandgap photoinduced absorption feature at 1.405 eV that decays very quickly and likely arises from hot carrier cooling or bandgap renormalisation effects[31–33]. The KK analysis, therefore, allows us access to broadband information that would otherwise only be available through transient transmission experiments ultrathin GaAs wafers grown on transparent substrates.

## Discussion

From the three instances of experimental pump-probe spectra discussed, it is clear that spectral signatures of both $\Delta\varepsilon_1$ and $\Delta\varepsilon_2$ can imprint on transient spectra despite $\Delta\varepsilon_2$ being the property of experimental interest. The relative contributions of $\Delta\varepsilon_{1,2}$ to the differential transmission in CsPbBr$_3$ and pentacene can be understood as follows. In the simplest scenario, ignoring Fabry–Pérot resonances, the thin film transmission $T = |(1 - r^2)t|^2$ where t and r and the reflectance and transmittance coefficients for the incident electric field. The total transmitted intensity is therefore modulated by the reflection from the front and back surface of the thin film due to refractive index mismatches at the interface. Below the bandgap where $t$ is close to 1, it is changes in the reflectivity that arises from changes in the refractive index that appear to dominate modulations in the transmitted light. In pentacene, while the refractive index of n ≈ 1.5 is much closer to that of air (n = 1), giving rise to a smaller overall reflected intensity, the modulation in the differential transmission arising from the reflection below the bandgap is relatively large[28]. For CsPbBr$_3$ despite the larger refractive index of n ≈ 1.8 causing more totally reflected light, as the perovskite absorbs more light, the modulations in the differential transmission coming from the reflection is smaller[34]. Changes in the refractive index are significant in both thin-film samples, however, their effect on the measured differential transmission is different and difficult to anticipate without the KK analysis discussed here. Further, in GaAs the transient reflection cannot be directly interpreted without aid of the KK analysis. Finally, in more involved transmission functions for multilayer samples that can be calculated using transfer matrix modelling, the effect of a photoexcitation on the effective optical constants require more careful, layer by layer analysis[18].

Importantly, the KK analysis has demonstrated the capacity to quantitively extract changes in $\Delta\varepsilon_2$. This could allow for the deconvolution of bulk and surface photophysics in thin-film semiconductors by comparing $\Delta\varepsilon_2$ extracted from the top surface transient reflection and the transient transmission. In organic materials such a quantitative analysis could be applied to access to the triplet population. This lays the foundation for further possibilities of calculating the singlet fission efficiency accurately without the need for triplet sensitisation experiments commonly used to estimate this quantity[35,36]. Further, given the excellent precision with which the band structure of GaAs is understood, quantitative extraction of $\Delta\varepsilon_2$ offers an direct link to properties calculable through first-principles density functional theory in order to build a fs-time resolved picture of the band structure and carrier occupations upon photoexcitation[37,38]. This approach could also give insights and clarify the attribution of spectral signatures to the optical Stark effect in semiconductors[39].

While we have used the method described here to study thinfilm semiconductor transmission and reflection from a surface of a bulk semiconductor, the extension to more complex samples, such as 2D excitonic systems or multilayer structures with resonances is straightforward and only requires a change in the spectral function $S(\tilde{\varepsilon}(\omega))$ and the calculation of the derivatives, typically facilitated by Wolfram Mathematica[3]. In order to facilitate the widespread implementation of our approach, we provide a Jupyter notebook Python code for reflection and transmission which is underlies all the data analysis in this paper which we have verified against results obtained using the Reffit software[14]. As this technique requires no additional experimental input other than the ground state and transient spectra it also facilitates retrospective analysis on already acquired pump-probe data sets, without the need to remeasure them.

Finally, we remark that, while powerful, the KK-based differential dielectric function has certain limitations to its scope. First,

the perturbative approach taken here does not apply at very high photoexcitation densities or for very large spectral shifts in the underlying absorption, where the approximation in Eq. 5 fails and more terms in the Taylor expansion are be required. Second, the KK constraint becomes less accurate when the broadband spectrum does not contain all the key spectral features, for example in the case of a strong and broad photoinduced band outside the probe spectral range in the deep IR caused by a visible pump.

In summary, we have discussed and demonstrated the use of a KK constrained differential dielectric function with variational oscillators to retrieve the full changes in the dielectric function from broadband pump-probe spectroscopy. This allows for complete quantitative access to the transient JDOS, removes ambiguities in the interpretation of spectral line shapes, and improves upon the absorption coefficient approximation to the imaginary part of the dielectric function. These methods have been applied to materials where the differential transmission and reflection spectra are very different and could be applied to any spectral function $S(\widetilde{\varepsilon}(\omega))$ so long as the derivatives $\frac{\partial S}{\partial \varepsilon_{1,2}}$ are analytically calculated. The extraction of the transient dielectric function from pump-probe spectroscopy forms a robust framework with which to repeatably and reliably interpret these experiments free of optical artefacts and with true quantitative precision.

## Methods

**Fabrication of a thin film of pentacene**. Thin films of pentacene were prepared by thermal evaporation in an ultrahigh vacuum environment of $10^{-8}$ mbar at a constant evaporation rate of 0.02 nm/s. The rate was monitored using a calibrated quartz crystal microbalance and the deposition was stopped once the desired film thickness of 100 nm was obtained. The deposition was done onto 1 mm-thick quartz substrates and 0.2 mm-thick glass coverslips that were cleaned by sequential sonication in acetone and isopropyl alcohol.

**Broadband pump-probe spectroscopy of pentacene**. A pump-probe measurements were performed using a homebuilt setup around a Yb:KGW amplifier laser (1030 nm, 38 kHz, 15 W, Pharos, LightConversion). The probe pulse was a chirped seeded white light continuum created using a 4 mm Yag crystal that spanned from 500 to 950 nm. The pump pulse was created using a non-collinear optical parametric amplifier (NOPA) where the 1030 nm seeded a white light continuum stage in sapphire which was subsequently amplified with the third harmonic of the 1030 nm laser in a beta barium borate crystal to create a broad pulse centred at 550 nm. This pulse was compressed using a chirped mirror and wedge prism (Layerterc) combination to a temporal duration of 18 fs. The average fluence of the pump 10 μJ/cm².

**Broadband pump-probe spectroscopy of GaAs**. A pump-probe measurements were performed using a homebuilt setup around a Yb:KGW amplifier laser (1030 nm, 38 kHz, 15 W, Pharos, LightConversion). The probe pulse was a chirped seeded white light continuum created using a 4 mm Yag crystal that spanned from 500 to 950 nm. The pump pulse was created using a NOPA where the 1030 nm seeded a white light continuum stage in Yag which was subsequently amplified with the second harmonic of the 1030 nm laser in a beta barium borate crystal to create a broad pulse centred at 700 nm. This pulse was compressed using a chirped mirror and wedge prism (Layerterc) combination to a temporal duration of 14 fs which was characterised using a homebuilt frequency resolved optical gating (FROG). The average fluence of the pump was 5 μJ/cm².

**Time resolved frequency domain interferometry experiment**. The FDI measurements on the Pentacene sample is performed using a home-built FDI setup. The driving laser is a Ti:sapphire regenerative amplifier system (805 nm, 3 kHz, 3W, Spitfire, SpectraPhysics). The broadband pump and reference-probe pair are generated by a double-staged Multiple Plate Compression technique, spanning from 500 to 950 nm[7,40]. The first back reflection of a wedge pair is used for the reference and probe arm while the transmitted beam is used as a pump. A 750 nm short-pass filter is used in the pump path to eliminate the strong fundamental wavelengths. An interferometer setup is used to create a time delay between the reference and probe pulse. The reference and probe are focused to a Gaussian profile with a diameter of 50 μm FWHM and the pump is focused to a Gaussian profile with a diameter of 350 μm FWHM. The spectrally integrated pump energy is 400 μJ/cm² per pulse.

## Data availability

The data that support the plots within this paper and other findings of this study are available at the University of Cambridge Repository (https://doi.org/10.17863/CAM.80129).

## Code availability

The code used in the analysis can be found on the Github Repository (link: https://github.com/arjun-ashoka/KKR_Spectral_Analysis, https://doi.org/10.5281/zenodo.5864960) as a Python Jupyter notebook with the corresponding datasets used in all the analysis for this paper.

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

## Acknowledgements

A.A. acknowledges funding from the Gates Cambridge Trust and as well as support from the Winton Programme for the Physics of Sustainability. A.V.G. acknowledges funding from the European Research Council Studentship and Trinity-Henry Barlow Scholarship. M.B.P. acknowledges support from the MacDiarmid Institute for Advanced Materials and Nanotechnology. C.S. acknowledges financial support by the Royal Commission of the Exhibition of 1851. We acknowledge financial support from the EPSRC and the Winton Programme for the Physics of Sustainability. This project has received funding from the European Research Council (ERC) under the European Union's Horizon 2020 research and innovation programme (grant agreement no. 758826).

## Author contributions

A.A. and A.R. conceived the idea. A.R. planned and supervised the project. R.R.T., S.Y., C.H., J.M.H., M.B.P., and K.C. designed and conducted the FDI experiments. A.V.G. prepared the pentacene sample. C.C., D.R., and C.G.S. provided the GaAs sample. S.D.V conducted the broadband pump-probe measurements on pentacene. A.A. and C.S. conducted the broadband pump-probe measurements on GaAs. A.A. developed the KK-based analysis and analysed all the data. All authors discussed the results and contributed to writing the manuscript.

## Competing interests

The authors declare no competing interests.
