## [Peer Review File · Nature Communications]

REVIEWER COMMENTS

Reviewer #1 (Remarks to the Author):

The paper by Rao and coworkers presents a novel approach for the analysis of ultrafast pump-probe experiments, in order to quantitatively recover the complex dielectric function $\epsilon = \epsilon_1 + i\epsilon_2$ based on a constrained Kramers-Kronig analysis. The basic idea of the approach is as follows: in pump-probe one is typically interested to measure the absorption changes (transient absorption, TA) but what one experimentally measures are the changes in transmission (differential transmission, DT/T) which contain also changes in the sample reflectivity at the two interfaces with the surroundings. In some cases these reflectivity changes may dominate the signal and be mistakenly interpreted as TA signals. It is therefore important to extract the true TA spectra from the DT/T spectra. This paper proposes an approach in which the experimental data are fitted in terms of variations of both the real and the imaginary part of the dielectric function, with a Kramers-Kronig constraint imposed between them. To obtain the dielectric function, a recently proposed approach is employed making use of a large set of Lorentzian functions which are devoid of physical meaning.

The analytic approach proposed in this paper is applied to a number of samples (perovskite thin films, pentacene films, GaAs wafer) and the results are in good agreement with independent measurements of the variation of ϵ_1 using spectral interferometry. Overall I find this paper well written and the method useful and of general applicability, so that it could have an impact on the broad community using ultrafast optical spectroscopy to study a variety of problems in physics and materials science.

Before publication, I have the following remarks:

- 1) it is not clear if a transmission transfer matrix (TTM) approach is used to calculate the sample transmission: this will become very relevant for films of samples, such as perovskites, which have thicknesses comparable to the wavelength and for which Fabry-Perot interference effects become relevant. It should be clarified in a revised version of the manuscript.
- 2) the authors mix the languages of semiconductor physics (valence band, conduction band) and molecular physics (ground state, excited state) as for example in page 4 in which they refer to ground state and excited state transitions. I would suggest to use a consistent language throughout the manuscript.
- 3) on page 4 the authors state that "while this formalism does not include stimulated emission (SE) features, with appropriate interpretation of spectral photobleaching features one can deconvolve the SE band from the true transient JDOS." This statement is not clear. Why does the model not include stimulated emission and how can it be recovered?

Reviewer #2 (Remarks to the Author):

This is a great paper. The topic has not been studied before in the present context, likely because KK relationships are not strictly possible for 3rd order spectroscopies. However, it is true that if one separates the pump from the probe, then a KK relationship for the probe can be formulated. Hence the pump-induced change of dielectric properties can be detected. The results should give insight into phenomena such as the optical Stark effect. Some nice examples are provided in the paper. I think it can be published in the present form.

Reviewer #3 (Remarks to the Author):

The authors correctly state the problem that transient transmission and reflection experiments are difficult to interpret, because the observed changes are due to transient effects in the COMPLEX dielectric function. A simple transmission or reflection experiment is unable to distinguish between the real and imaginary parts of the dielectric function, because both affect the experiment.

The authors therefore suggest a Kramers-Kronig technique to determine the transient dielectric function from the measurement transmission or reflection spectral. This is not novel, because the KK technique has been around for a long time. It is also flawed, because changes in the dielectric function are NOT local in energy space. For example, an infrared pump pulse can have significant changes in the visible dielectric function.

Finally, this numerical analysis is not needed, because other techniques are available to determine the transient dielectric function directly without any KK analysis. For example, Mazur (Harvard) used pump-probe reflectance at multiple angles of incidence. Shank and Auston used pump-probe ellipsometry. The latter technique is now available at an international user facility to a broad range of researchers (ELI Beamlines, Dolni Brezany, Czech Republic).

In my opinion, the technique proposed by the authors is not noteworthy. It is also flawed and only needed for obsolete optical pump-probe techniques, such as single-angle reflection or transmission.

Authors' Response to Referees' Comments: NCOMMS-21-35848

We thank the three referees for carefully going through our manuscript and for their valuable comments and suggestions. In light of their comments we have made revisions to the manuscript and supplementary information (highlighted in yellow).

Point-wise responses:

Reviewer #1 (Remarks to the Author):

“The paper by Rao and coworkers presents a novel approach for the analysis of ultrafast pump-probe experiments, in order to quantitatively recover the complex dielectric function $\varepsilon = \varepsilon_1 + i \varepsilon_2$ based on a constrained Kramers-Kronig analysis. The basic idea of the approach is as follows: in pump-probe one is typically interested to measure the absorption changes (transient absorption, TA) but what one experimentally measures are the changes in transmission (differential transmission, DT/T) which contain also changes in the sample reflectivity at the two interfaces with the surroundings. In some cases these reflectivity changes may dominate the signal and be mistakenly interpreted as TA signals. It is therefore important to extract the true TA spectra from the DT/T spectra. This paper proposes an approach in which the experimental data are fitted in terms of variations of both the real and the imaginary part of the dielectric function, with a Kramers-Kronig constraint imposed between them. To obtain the dielectric function, a recently proposed approach is employed making use of a large set of Lorentzian functions which are devoid of physical meaning. The analytic approach proposed in this paper is applied to a number of samples (perovskite thin films, pentacene films, GaAs wafer) and the results are in good agreement with independent measurements of the variation of ε_1 using spectral interferometry. Overall I find this paper well written and the method useful and of general applicability, so that it could have a impact on the broad community using ultrafast optical spectroscopy to study a variety of problems in physics and materials science.”

We thank the reviewer for their concise summary of the manuscript, for restating the importance of such analysis to extract the true photoexcited dynamics and spectra in pump-probe measurements and for supporting that this paper can have an impact on the broad community using pump-probe spectroscopy.

“Before publication, I have the following remarks:

1) it is not clear if a transmission transfer matrix (TTM) approach is used to calculate the sample transmission: this will become very relevant for films of samples, such as perovskites, which have thicknesses comparable to the wavelength and for which Fabry-Perot interference effects become relevant. It should be clarified in a revised version of the manuscript.”

We thank the reviewer for highlighting this important point. Throughout the manuscript we used Fresnel coefficients directly, which is equivalent to a transmission transfer matrix (TTM) approach for a simple, bifacial thin film in the incoherent limit. As the reviewer has rightly pointed out, for some sample geometries a TTM approach is better suited to capture the relevant optical constants. We emphasise that, so long as the overall transmission function $T(\varepsilon_1, \varepsilon_2)$ is calculated, be it through the TTM model or other situations (such as 2D material's transmission function), the methodology we have developed can be applied. **To clarify this we have:**

a) Modified paragraph 2 on page 5 and referred the reader to citation [18] where these effects are studied in more details:

“We use the functional form of $T(\epsilon_1, \epsilon_2)$ for a bifacial thin film geometry using Fresnel Equations (see SI 2). We note that for more complex sample structures including interferences and Fabry–Pérot resonances, $T(\epsilon_1, \epsilon_2)$ can be efficiently calculated using transfer matrix modelling^{1,2}. Once $T(\epsilon_1, \epsilon_2)$ has been parameterised, it is possible to calculate the derivatives $\left(\frac{\partial T}{\partial \epsilon_1}\right)$ and $\left(\frac{\partial T}{\partial \epsilon_2}\right)$ analytically as a function of ϵ_1 and ϵ_2 .”

b) Added the following statement to paragraph 1 on page 10:

“Finally, in more involved transmission functions for multilayer samples that can be calculated using transfer matrix modelling, the effect of a photoexcitation on the effective optical constants require more careful, layer by layer analysis.²”

c) Modified paragraph 2 in the Supplementary Information section SI 2:

“We note that while we have ignored the interference based Fabry–Pérot resonances, similar expression for the transmission and reflection for a multilayer sample can be calculated in both the coherent and incoherent regimes using the transfer matrix method.¹”

“2) the authors mix the languages of semiconductor physics (valence band, conduction band) and molecular physics (ground state, excited state) as for example in page 4 in which they refer to ground state and excited state transitions. I would suggest to use a consistent language throughout the manuscript.”

We thank the reviewer for pointing this out. We have now adopted the use of the more general language of excited and ground state and only use the phrase valence and conduction band as examples of the terms appearing in Eqn 2.

We have therefore modified paragraph 2 and 3 on page 3 as follows:

“For a material subject to an oscillating transverse electromagnetic field, using Fermi’s golden rule and the macroscopic power dissipation theorem, ϵ_2 is given as,

$$\epsilon_2(\omega) = \frac{2\pi}{N_e m_e} \sum_{i \in GS} \frac{\omega_p^2}{\omega_i^2} \sum_T \underbrace{|\mathbf{p}_{Ti}|^2}_{\text{Oscillator Strength}} \underbrace{\delta(E_{Ti} - \hbar\omega)}_{\text{Joint Density of States (JDOS)}} \quad (2)$$

where the real part of the dielectric function can be obtained by the Kramers-Kronig transform,

$$\epsilon_1(\omega) = \frac{1}{\pi} \mathcal{P} \int_{-\infty}^{\infty} \frac{\epsilon_2(\omega')}{\omega' - \omega} d\omega'. \quad (3)$$

Here, m_e is the electron mass, N_e is the number of electrons in molecular or crystal unit cell, ω_p is the plasma frequency, ω the photon frequency and the factor of 2 accounts for the spin degeneracy. The first sum (over i) extends over the available states in the ground state (GS) manifold (or equivalently the Brillouin Zone for a crystal) and the second sum is over the available transitions (T). The matrix elements of the momentum operator between the wavefunctions in the ground and excited (or valence and conduction bands for a crystal) is

given as p_{T_i} with an associated transition energy E_{T_i} . The matrix elements of the momentum operator can be recast as the electric dipole operator that represents the oscillator strength of the transition. The Dirac delta function represents the existence of states in the ground and excited state manifold that would allow for a transition at an energy $\hbar\omega$, which is referred to as the joint density of states (JDOS). This picture is naturally modified for excitonic systems, but the overall dependence of ϵ_2 on the joint density of states and the dipole matrix elements remains universal³."

3) one page 4 the authors state that "while this formalism does not include stimulated emission (SE) features, with appropriate interpretation of spectral photobleaching features one can deconvolve the SE band from the true transient JDOS." This statement is not clear. Why does the model not include stimulated emission and how can it be recovered?

Our model does not include the stimulated emission (SE) band as it focuses solely on constructing the photoexcited weighted density of states in the material. Just like photoluminescence, which is also not included in this model, changes in the net transmission that result from radiation from the material rather than changes in optical constants are not explicitly captured in our analysis. This sentence was intended to highlight this caveat.

In order to fully recover the stimulated emission band from this material, there exist two options depending on the magnitude of the Stokes shift. If the Stokes shift is large, one can study the photobleaching-like negative spectral signatures in $\Delta\epsilon_2$ that appear shifted from the expected ground state photobleaching feature (based on the static ϵ_2), just as is typically done in transient absorption spectroscopy. However, if the Stokes shift is small, one can combine the methodology we provide to calculate $\Delta\epsilon_{2_KKR}$ that includes the SE band with the Frequency Domain Interferometry (FDI) retrieved $\Delta\epsilon_{2_FDI}$ which is sensitive *solely* to the real part of the refractive index and therefore does not pick up the SE band. The spectrum of $\Delta\epsilon_{2_KKR} - \Delta\epsilon_{2_FDI}$ should contain purely the SE band. Such an approach could resolve previous uncertainty in transient transmission measurements where the Stokes shift is small and overlaps with the ground state bleach spectrum, obscuring the stimulated emission band and lifetime.

To clarify this, we have added the following statement to paragraph 3, page 4:

"We note that this formalism does not include stimulated emission (SE) features as net transmission changes that result from radiation from the material rather than changes in optical constants do measure purely the transient photoexcited JDOS as they do not appear in Eqn 2. However, with appropriate interpretation of spectral photobleaching features in $\Delta\epsilon_2(\omega)$, based on the magnitude of the Stokes shift, one can deconvolve the SE band from the true transient JDOS (SI 1).

In addition, we now provide a more detailed description in a new section of the Supplementary Information (SI 1), in the hope to be further explain these points.

Reviewer #2 (Remarks to the Author):

“This is a great paper. The topic has not been studied before in the present context, likely because KK relationships are not strictly possible for 3rd order spectroscopies.”

We thank the reviewer for their enthusiasm in the paper and for their insightful point that the KK relationship is usually not possible in 3rd order spectroscopies.

We have now included this detail in the manuscript in paragraph 4 on page 4:

“We note that while usually the KK relations are not strictly obeyed in third-order spectroscopies, due to the strict time ordering of the pump and probe pulse, the KK relation can be applied to the probe spectrum.”

However, it is true that if one separates the pump from the probe, then a KK relationship for the probe can be formulated. Hence the pump-induced change of dielectric properties can be detected. The results should give insight into phenomena such as the optical Stark effect. Some nice examples are provided in the paper. I think it can be published in the present form.”

We thank the reviewer for their concise summary of the paper and support.

We have now also included a sentence and citation [40] stating the potential use of the technique to study the optical Stark effect in the paper in paragraph 2 on page 10:

“This approach could also give insights and clarify the attribution of spectral signatures to the optical Stark effect in semiconductors⁴.”

Reviewer #3 (Remarks to the Author):

“The authors correctly state the problem that transient transmission and reflection experiments are difficult to interpret, because the observed changes are due to transient effects in the COMPLEX dielectric function. A simple transmission or reflection experiment is unable to distinguish between the real and imaginary parts of the dielectric function, because both affect the experiment.”

We thank the reviewer for their concise summary of the manuscript and for restating the importance of extracting the true transient dielectric properties from a pump-probe experiment, which is the central premise of our manuscript.

The authors therefore suggest a Kramers-Kronig technique to determine the transient dielectric function from the measurement transmission or reflection spectral. This is not novel, because the KK technique has been around for a long time.

We agree with the reviewer that forms of the Kramers-Kronig (KK) technique applied to optical spectroscopy have been around since the original papers by Kramers and Kronig in 1925/1927. In fact, these have informed and motivated our study and are cited at the relevant sections, and it was never our intention to portray the KK methodology as novel. Instead, our work offers a complete and self-consistent analytical KK framework to adequately interpret nowadays routine pump-probe spectroscopies by considering the pump-induced changes to the complex dielectric function in materials. We rigorously benchmarked our approach for different sample geometries and material compositions against independent FDI measurements and packaged it in an open-source Python code, to be as widely and easily accessible as possible. We firmly believe this toolkit will serve as an invaluable resource to the large community employing pump-probe spectroscopies.

We have now modified paragraph 5 on page 2 to better reflect this:

“Here, we present and benchmark a model independent and universally applicable route to transform broadband differential optical spectra to changes in the real and imaginary part of the dielectric function from which changes in the true underlying excitation spectrum of the material can be confidently assigned. We make use of the well-established Kramers-Kronig (KK) relations and their variational counterpart⁵⁻⁷.”

“It is also flawed, because changes in the dielectric function are NOT local in energy space. For example, an infrared pump pulse can have significant changes in the visible dielectric function.”

The reviewer makes valid point, which we believe is referring to the following statement of our original manuscript in **paragraph 4 on page 4**:

“Critically, however, $\Delta\tilde{\epsilon}(\omega)$ can be considered as an energetically local perturbation to the dielectric function and hence most of the features, especially in the weak photoexcitation limit, are also energetically local and the KK constraint can be applied over the narrower energy range where the spectral features are present.”

We have taken the reviewers concerns on board and **revised this paragraph** in the hope that it will clarify the point the reviewer raised:

“A well-established complication arising from the use of the KK relation on the ground state dielectric function is that experimental limitations truncate the frequency range that can be

accessed which causes use of the KK relations on the measured dielectric properties to become less exact. The effect of transitions a distance $\omega - \omega_0$ from a transition at ω_0 are however, suppressed by the term $1/(\omega - \omega_0)$ in the KK relations (Eqn 3) and hence, while non-locality is an ever-present issue with use of the KK relations, they can still be applied over finite experimental bandwidths⁸. In typical semiconducting materials however, $\Delta\tilde{\epsilon}(\omega)$ is usually an energetically local perturbation to the dielectric function and hence most of the features, especially in the weak photoexcitation limit, are also energetically local and the KK constraint can be accurately applied over the narrower energy range where the spectral features are present.”

While we agree that truncating the energetic range studied to a few eV is indeed an approximation that we are making, this is precisely why we benchmark our extracted real refractive index (which the transient transmission experiment is only weakly sensitive to) to the values extracted independently from Frequency Domain Interferometry (FDI). The quantitative agreement that we find on both simple (CsPbBr₃) and congested (Pentacene) spectra is evidence that our assumption is justified in most cases under the conditions laid out in the manuscript.

To bring out this point more clearly, we have added the following statement to paragraph 1, page 7:

“As illustrated in Figure 2d, we find quantitative agreement between our calculated $\Delta n(\omega)$ spectra obtained via variational KK analysis (solid lines) and the measured spectra (open circles), validating our approach and the use of the KK relations on a relatively energetically narrow spectrum.”

We have also added the following statement to paragraph 3, page 8:

“This demonstrates that the variational KK analysis can be applied on congested pump-probe spectra and achieves quantitative precision on the signal shape and peak to peak ratios further validating our approach and the use of the KK relations on a relatively energetically narrow spectrum.”

In our final paragraph of the manuscript (paragraph 4, page 10) we have explicitly taken note of the issue of very broadband spectra where we find that the model becomes less accurate:

“Second, the KK constraint becomes less accurate when the broadband spectrum does not contain all the key spectral features, for example in the case of a strong and broad photoinduced band outside the probe spectral range in the deep IR caused by a visible pump.”

“Finally, this numerical analysis is not needed, because other techniques are available to determine the transient dielectric function directly without any KK analysis. For example, Mazur (Harvard) used pump-probe reflectance at multiple angles of incidence. Shank and Auston used pump-probe ellipsometry. The latter technique is now available at an international user facility to a broad range of researchers (ELI Beamlines, Dolni Brezany, Czech Republic).”

We respectfully disagree with reviewer on the premise that a numerical analysis technique is not needed just because other, more complicated experimental techniques already exist. For example, we would like to emphasize that our numerical analysis can be used retrospectively on already acquired pump-probe data sets, without the need to remeasure them. We believe this

is a unique advantage of the KK analysis technique and will allow for the community to clarify historical ambiguities in the origin of certain spectral features. **We have now explicitly highlighted this point in the manuscript in paragraph 3, page 10:**

“As this technique requires no additional experimental input other than the ground state and transient spectra it also facilitates retrospective analysis on already acquired pump-probe data sets, without the need to remeasure them.”

We thank the reviewer for highlighting some very important experimental techniques. These techniques, while powerful, suffer from complicated and specialist setups which exist in very few labs across the work and hence have limited access by the community. As stated by the reviewer, some of these are user facilities, rather than widely available and standard instruments. In stark contrast, conventional pump probe spectroscopy is nowadays a routine measurement tool available in thousands of academic and industrial research settings. Given the wide and growing use of pump-probe spectroscopy, we consider that our methodology, which allows new information to be gained from the method will be very useful to a broad community of users.

We, however, agree with the reviewer that these techniques they mention should be mentioned in context, which is **why we have now included them explicitly (references [9] and [10]) and have clarified why these measurements are typically challenging on thin film materials in paragraph 4 on page 2:**

“Initial approaches to mitigate these ambiguities in studying differential spectra have considered measuring both the differential transmission and reflection, multiangle reflectance or the optical ellipticity from which the true photoexcited transient absorption changes can be calculated^{9,10}. These methods are however, experimentally challenging on the commonly studied thin film materials and not universally applicable, especially on solution processed thin film semiconductors where morphological microstructures complicate the analysis¹¹.”

Notably, in contrast to these experimental techniques our analysis technique provides an easier, faster solution that can be used to complement these measurements, potentially even initially screening material spectra measured in standard pump-probe labs to aid in the selection of the correct facility.

Finally, with regards to the reviewers previous point about the non-locality of the spectral signatures, we note that these facilities suffer from the same issues as broadband pump probe, as they too truncate the spectral range accessible and apply then apply the KK relation. The expectation that one must measure out to 10s of eV and down to meV ranges for such measurements is definitely the ultimate goal for pump-probe spectroscopy but this is not yet within the experimental realm of possibility.

“In my opinion, the technique proposed by the authors is not noteworthy. It is also flawed and only needed for obsolete optical pump-probe techniques, such as single-angle reflection or transmission.”

We respectfully disagree with the reviewer’s statement that single-angle of incidence, transmission only, pump-probe measurements are an obsolete technique. Pump-probe spectroscopy is nowadays a routine measurement tool available in thousands of academic and industrial research settings. This ubiquity is itself a great power of this technique and therefore

while there might be more complicated experimental techniques available, they do not make pump-probe spectroscopy obsolete. Just as conventional Raman spectroscopy is not made obsolete via the invention of Surface Enhanced Raman or Time-Resolved Raman methodologies, so it is that pump-probe spectroscopy continues to be a mainstay of the community. Given this wide and growing use of pump-probe spectroscopy, we consider that our methodology, which allows new information to be gained from the method will be very useful to a broad community of users.

We point to the a small selection of papers published in and after 2020 in Nature and Science which utilise this technique^{12–20}. This account further supports our previous claims, that pump-probe spectroscopy is nowadays routinely used, but often incorrectly interpreted. Our manuscript aims to provide that community with the necessary analytical tools to avoid misinterpretation.

References

1. Troparevsky, M. C., Sabau, A. S., Lupini, A. R. & Zhang, Z. Transfer-matrix formalism for the calculation of optical response in multilayer systems: from coherent to incoherent interference. *Opt. Express* **18**, 24715 (2010).
2. Renken, S. *et al.* Untargeted Effects in Organic Exciton-Polariton Transient Spectroscopy: A Cautionary Tale. **154701**, (2021).
3. Alvertis, A. M. *et al.* First principles modeling of exciton-polaritons in polydiacetylene chains. *J. Chem. Phys.* **153**, 1–14 (2020).
4. Fröhlich, D., Nöthe, A. & Reimann, K. Observation of the Resonant Optical Stark Effect in a Semiconductor. *Phys. Rev. Lett.* **55**, 1335–1337 (1985).
5. Kramers, M. H. A. La diffusion de la lumière par les atomes. *Trans. Volta Centen. Congr* **2**, 545–557 (1927).
6. Kronig, R. Optical Society of America Review of Scientific Instruments. *J. Opt. Soc. Am.* **12**, 459–463 (1925).
7. Kuzmenko, A. B. Kramers-Kronig constrained variational analysis of optical spectra. *Rev. Sci. Instrum.* **76**, 1–9 (2005).
8. Milton, G. W., Eyre, D. J. & Chen, B. Finite frequency range kramers-kronig relations: Bounds on the dispersion. *Phys. Rev. Lett.* **79**, 3062–3065 (1997).
9. Roeser, C. A. D. *et al.* Femtosecond time-resolved dielectric function measurements by dual-angle reflectometry. *Rev. Sci. Instrum.* **74**, 3413–3422 (2003).
10. Auston, D. H. & Shank, C. V. Picosecond Ellipsometry of Transient Electron-Hole Plasmas in Germanium. *Phys. Rev. Lett.* **32**, 1120–1123 (1974).
11. Liu, J., Leng, J., Wang, S., Zhang, J. & Jin, S. Artifacts in Transient Absorption Measurements of Perovskite Films Induced by Transient Reflection from Morphological Microstructures. *J. Phys. Chem. Lett.* **10**, 97–101 (2019).
12. D., S. *et al.* Mechanism and dynamics of fatty acid photodecarboxylase. *Science (80-.)*. **372**, eabd5687 (2021).

13. Gillett, A. J. *et al.* The role of charge recombination to triplet excitons in organic solar cells. *Nature* **597**, 666–671 (2021).
14. Han, S. *et al.* Lanthanide-doped inorganic nanoparticles turn molecular triplet excitons bright. *Nature* **587**, 594–599 (2020).
15. Li, Y. *et al.* Double-helical assembly of heterodimeric nanoclusters into supercrystals. *Nature* **594**, 380–384 (2021).
16. Xu, J. *et al.* Magnetic sensitivity of cryptochrome 4 from a migratory songbird. *Nature* **594**, 535–540 (2021).
17. Yi, H. *et al.* Efficient tandem solar cells with solution-processed perovskite on textured crystalline silicon. *Science (80-.)*. **367**, 1135–1140 (2020).
18. Young-Hoon, K. *et al.* Chiral-induced spin selectivity enables a room-temperature spin light-emitting diode. *Science (80-.)*. **371**, 1129–1133 (2021).
19. Hassan, Y. *et al.* Ligand-engineered bandgap stability in mixed-halide perovskite LEDs. *Nature* **591**, 72–77 (2021).
20. MacKenzie, I. A. *et al.* Discovery and characterization of an acridine radical photoreductant. *Nature* **580**, 76–80 (2020).

REVIEWERS' COMMENTS

Reviewer #1 (Remarks to the Author):

The revised version of the manuscript fully addresses the concerns raised by me and by the other reviewers. The paper is now ready for publication in Nature Communications.

Authors' Response to Referees' Comments: NCOMMS-21-35848

We thank the three referees for carefully going through our manuscript and for their valuable comments and suggestions.

Point-wise responses:

Reviewer #1 (Remarks to the Author):

“The revised version of the manuscript fully addresses the concerns raised by me and by the other reviewers. The paper is now ready for publication in Nature Communications.”

We thank the reviewer for their positive endorsement for publication.